# Massive yet grossly underestimated global costs of invasive insects

Corey J.A. Bradshaw[1,2], Boris Leroy[1,3], Céline Bellard[1,4], David Roiz[5,*], Céline Albert[1,*], Alice Fournier[1], Morgane Barbet-Massin[1], Jean-Michel Salles[6], Frédéric Simard[5] & Franck Courchamp[1,7,8]

Insects have presented human society with some of its greatest development challenges by spreading diseases, consuming crops and damaging infrastructure. Despite the massive human and financial toll of invasive insects, cost estimates of their impacts remain sporadic, spatially incomplete and of questionable quality. Here we compile a comprehensive database of economic costs of invasive insects. Taking all reported goods and service estimates, invasive insects cost a minimum of US$70.0 billion per year globally, while associated health costs exceed US$6.9 billion per year. Total costs rise as the number of estimate increases, although many of the worst costs have already been estimated (especially those related to human health). A lack of dedicated studies, especially for reproducible goods and service estimates, implies gross underestimation of global costs. Global warming as a consequence of climate change, rising human population densities and intensifying international trade will allow these costly insects to spread into new areas, but substantial savings could be achieved by increasing surveillance, containment and public awareness.

[1] Ecologie, Systématique et Evolution, Université Paris-Sud, CNRS, AgroParisTech, Université Paris-Saclay, 91400 Orsay, France. [2] School of Biological Sciences, University of Adelaide, Adelaide, South Australia 5005, Australia. [3] UMR 7208 Biologie des Organismes et des Ecosystémes Aquatiques, Muséum National d'Histoire Naturelle, Université Pierre et Marie Curie, Université de Caen Basse-Normandie, CNRS, IRD, Sorbonne Universités, 43 rue Cuvier, 75005 Paris, France. [4] Department of Genetics, Evolution and Environment, Centre for Biodiversity and Environment and Research, University College London, London WC1E 6BT, UK. [5] MIVEGEC, Université de Montpellier-IRD224-CNRS5290, 911 Avenue Agropolis, 34394 Montpellier, France. [6] Laboratoire Montpellierain d'Économie Théorique et Appliquée, Centre national de recherche scientifique, Institut national de recherche agronomique, SupAgro, Université de Montpellier, UPVM3, 34060 Montpellier, France. [7] Department of Ecology and Evolutionary Biology, University of California, 621 Young Drive South, Los Angeles, California 90095-1606, USA. [8] Center for Tropical Research, Institute of the Environment and Sustainability, La Kretz Hall, University of California Los Angeles, California 90095, USA. * These authors contributed equally to this work. Correspondence and requests for materials should be addressed to C.J.A.B. (email: corey.bradshaw@adelaide.edu.au) or to F.C. (email: franck.courchamp@u-psud.fr).

For millennia, insects have been responsible for spreading devastating infectious diseases in both humans[1] and livestock[2], ravaging crops and food stocks[3], damaging forests[4], destroying infrastructure[5], altering ecosystem functions[6] and weakening the resilience of ecosystems to other disturbances[7]. This single invertebrate class (~2.5 million species[8]) is therefore probably the costliest animal group to human society.

A global challenge this century will be meeting the world's food requirements while maintaining economic productivity and conserving biodiversity. Globally, insect pests have been reported to reduce agricultural yields by 10–16% before harvest, and to consume a similar amount following harvest[9]. In fact, the largest food-producing countries, China and the United States, exhibit the highest potential losses from invasive insects[10]. Several other insect pests defoliate trees[4] and degrade plant biodiversity, threaten commercial forestry and hamper climate change mitigation via increased tree mortality and associated increases in greenhouse-gas emissions[11]. Many other insects are nuisance species or disease vectors that directly erode public health—from the Seventeenth to Twentieth centuries, insect-borne diseases caused more human disease and death than all other causes combined[12].

Insects are also among the most pervasive of invasive species. For example, 87% of the ~2,500 non-native terrestrial invertebrates in Europe are insects[13]. Yet, reliable estimates of their impacts are difficult to obtain, in particular for economic assessments. Most cost estimates are disparate, regionally focused, cover variable periods and are not always grounded in verifiable data (see Methods). The types of costs also vary and include both direct and indirect components (Fig. 1). Consequently, extrapolating local costs to global scales is challenging and few have attempted to overcome the many inherent flaws in this approach.

Reliable global cost summaries therefore remain a major challenge. Indeed, there are currently only 86 insect species listed in the International Union for Conservation of Nature (IUCN) Global Invasive Species Database[14], and of those there are no cost estimates for 81.4%, while 12.8% of them have insufficient (unsourced) estimates. We therefore compiled the most comprehensive database of economic costs for invasive insects available to date (737 screened articles, chapters and reports), standardizing historical estimates as annual 2014 US dollars (US$; Methods).

We determined the reproducibility of each study's cost estimates by identifying the source of all values used to extrapolate regional costs. When values were based on actual measures as opposed to non-sourced estimates and had a clear methodology provided, we deemed the resulting costs 'reproducible' (although we did not assess quality per se because of a lack of standard, objective criteria to assess the accuracy of published estimates; Methods). We categorized studies that did not meet these criteria as 'irreproducible'. We further divided all costs into two main categories: 'goods and services' (including production of agricultural and forestry goods, and cultural services; Fig. 1) and 'human health', further splitting the former into agriculture, forestry, infrastructure, mixed or urban categories, and the latter into seven disease categories (Methods).

Taking all reported goods and services estimates, and avoiding the extrapolation of limited data, invasive insects cost a minimum of US$70.0 billion per year globally, while associated health costs exceed US$6.9 billion per year. Total costs rise as the number of estimates increases; therefore, the true costs of invasive insects to human society are substantially larger (but by a currently unquantifiable amount) than we report here. Further, future costs are likely to increase as invasive insects expand their ranges in response to climate change, as well as to increasing human movements and international trade.

## Results

**Goods and services.** We determined that invasive insects cost a minimum of US$70.0 billion per year globally for goods and services, of which US$25.2 billion per year comes from reproducible studies (Fig. 2 and Supplementary Data 1). There was no temporal pattern in annual cost rates (Supplementary Fig. 1), and most estimates were direct measures (although estimated costs were higher for extrapolated costs; see 'Expenditure types and targets' in the Supplementary Methods and Supplementary Fig. 2). Regionally, North America reported the highest annual costs (>US$27.3 billion), followed by Europe (US$3.6 billion per year; Fig. 2a,b), although this is likely more a function of the intensity of research effort (see 'Research effort' below) rather than a true reflection of relative regional costs. The 10 costliest species change little whether including all or only reproducible estimates (Fig. 2e,f).

According to a single study[5], the most expensive insect is purportedly the Formosan subterranean termite *Coptotermes formosanus* estimated at >US$30.2 billion per year globally (Fig. 2e). However, that irreproducible estimate is based on a single non-sourced value of US$2.2 billion per year for the United States of America, a personal communication supporting a ratio of 1:4 of control:repair costs in a single US city (New Orleans) and an unvalidated assumption that the US costs represent 50% of the global total[5]. A more realistic ranking based on the reproducible estimates only (Fig. 2f) places the diamondback moth *Plutella xylostella* as the most expensive (US$4.6 billion per year)[15]. Other costly insects include the brown spruce longhorn beetle *Tetropium fuscum* (US$4.5 billion per year in Canada), the gypsy moth *Lymantria dispar* (US$3.2 billion per year in North America) and the Asian long-horned beetle *Anoplophora glabripennis* (US$3.0 billion per year in North America and Europe; Fig. 2f).

**Human health.** Global health costs directly attributable to invasive insects exceed US$6.9 billion per year (Fig. 3); however, these exclude malaria costs because that disease is not due to the invasion of an insect vector throughout most of its distribution (although malaria cases 'imported' into non-endemic areas do incur treatment and prophylaxis costs[16]). Our summary also excludes the economic impacts on productivity, income, tourism, blood-supply system, personal protection and quality of life (Supplementary Note 1), as well as historical epidemics of yellow fever and dengue because no relevant cost estimates exist (Methods). Most health-related estimates are a combination of direct and indirect costs (79% and 93% for all estimates and reproducible-only estimates, respectively; see Supplementary Note 1), represent actual estimates as opposed to extrapolations or model predictions (66% and 77%, respectively) and are primarily related to medical care (75% and 88%, respectively; see 'Expenditure types and targets' in the Supplementary Methods and Supplementary Figs 3 and 4). Dengue (from a virus transmitted by *Aedes albopictus* and *Ae. aegypti*) costs represent 84% of total health costs, followed by 15% for West Nile virus transmitted by *Culex* spp. (Fig. 3c,d). Asia (US$2.84 billion) and North America (US$2.06 billion) and Central/South America (US$1.85 billion) recorded the highest annual health costs (Fig. 3a,b).

**Research effort.** The regional summaries for both goods and services and health costs belie a strong positive relationship between total costs and the number of individual estimates (see 'Sampling bias' in the Supplementary Methods and Supplementary Fig. 5). Across regions, goods and services costs increase by 10 times for each additional 5.5 (reproducible-only) or

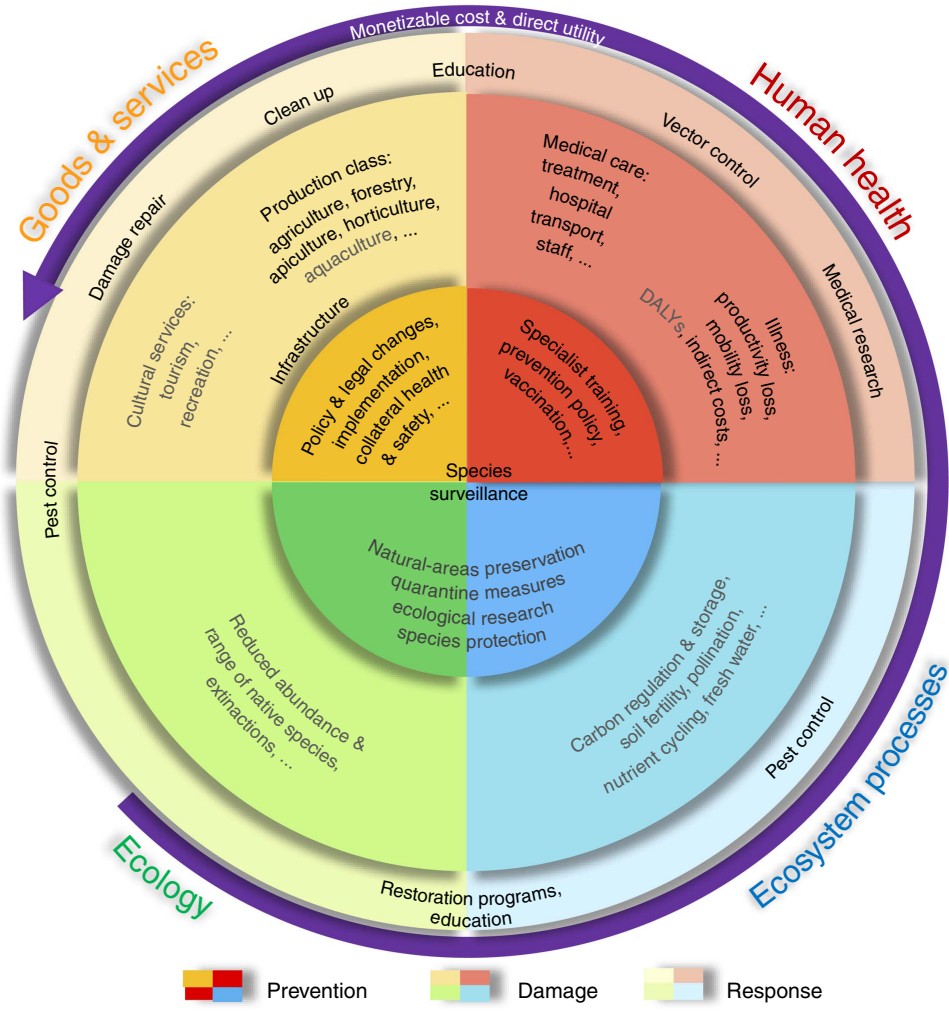

**Figure 1 | Market and non-market cost categories associated with invasive insect damages.** Costs are subdivided into 'goods and services' (yellow) and 'human health' (red), 'regulating services' (*sensu* non-commercial, but potentially monetizable, such as carbon regulation and pollination not otherwise quantified in agricultural yield estimates; blue) and 'ecological' costs (not typically monetizable; green). Owing mainly to a lack of monetary estimates, we could not compile costs for the categories and subcategories coloured in grey. The inner circle (darkest colours) encapsulates costs associated with prevention; the middle circle (mid-range colours) includes costs associated with damage from invasive insects; the outer circle (lightest colours) covers costs associated with responses or follow-up to invasive insect incursions. The outermost purple arrow indicates the general increase in our ability to estimate monetizable costs, and the direct relevance to human commerce and well being. DALY, disability-adjusted life year (lifespan lost because of burden of insect-borne disease; not assessed).

13.0 (all) estimates (Supplementary Fig. 5a,b). This strong positive relationship remains when expressed across species (Supplementary Fig. 6), but is necessarily more variable, given that most species have only one cost estimate each. The same type of relationship also exists for health costs, with total costs increasing by 10 times for each additional 18.5–19.1 estimates (Supplementary Fig. 6c,d). This regional bias in sampling corroborates the established phenomenon of a spatial mismatch between invader impacts on threatened species and research publications[17], suggesting that large additional costs because of invasive insects remain to be estimated in lesser-sampled regions of the world, and reinforcing our hypothesis that the total costs have been grossly underestimated.

**Cumulative costs**. Given that the regions to which these sums apply do not have the same spatial area, have different climates, have important crop and infrastructure differences, and are likely to experience different insect invasion and detection probabilities,

extrapolating regional costs to correct for potential undersampling is dubious. We therefore expressed total costs and the number of associated estimates as temporally cumulative values to identify possible thresholds within the sampled regions and categories (see 'Sampling bias' in the Supplementary Methods and Fig. 4). For both global goods and services and human health costs, there was evidence for an asymptote among the sampled species based on fitted logistic models (Fig. 4); however, reproducible-only goods and services costs had more support for a non-asymptotic linear model (Fig. 4b). This asymptotic behaviour is driven principally by North American goods and services costs (Supplementary Table 1 and Supplementary Fig. 7); in contrast, asymptotic behaviour was more prevalent across compared regions for human health costs (Supplementary Table 1 and Supplementary Fig. 8). For human health costs dominated by those associated with dengue fever, potential undersampling appears less problematic than for clearly underestimated costs from reproducible studies of goods and services. This variable asymptotic behaviour means that only some regions and cost

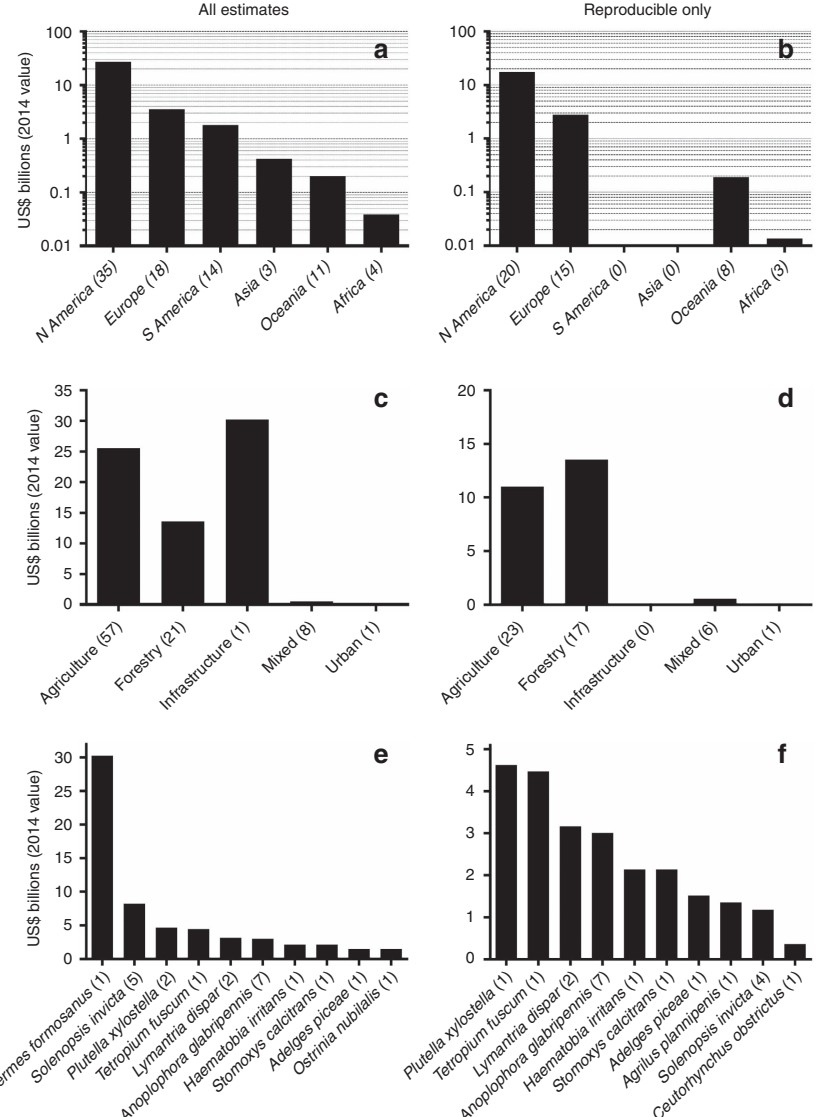

**Figure 2 | Goods and services costs associated with invasive insects.** Direct goods and services costs are categorized by major region (**a,b**), type (**c,d**) and by the 10 costliest insects (**e,f**). The first column includes all estimates regardless of reproducibility (**a,c,e**), whereas the second only includes costs for which estimates can be verified ('reproducible'; **b,d,f**). All costs expressed as annual 2014 US dollars. Bracketed numbers in the x axis labels indicate the number of estimates per category.

types demonstrate possible evidence of decelerating accumulation rates (that is, the costliest insects are assessed initially, with smaller damages estimated thereafter).

## Discussion

The estimated total global costs, even after attempting to correct for sampling bias, are therefore necessarily gross underestimates. We found only 86 (goods and services) and 117 (health) estimates globally, of which only 55% of the former ($n = 47$) and 85% of the latter ($n = 99$) we deemed reproducible. Ecosystem-regulating services, which have high economic value worldwide[18], are notoriously difficult to estimate[19]; hence, estimating the cost of their erosion arising from invasive insects is still unknown. In fact, we identified only one study[20] that provided reproducible economic costs of the erosion of ecosystem-regulating services (that is, costs not directly associated with goods and services or health, such as the erosion of pollination; Fig. 1) because of invasive insects (two *Vespula* wasps in New Zealand). That study

showed that damages arising mainly from reduced pollination are comparable to the direct costs to goods and services (for example, lost apicultural production and control) and are much higher than associated health costs[20].

While many non-native species are clearly beneficial to human society (Supplementary Fig. 9) by providing food, fibre, ecosystem services and even ecological benefits (habitats and resources for native species[21]; *ex situ* conservation[22]; increasing reproductive success of native plants[23]), the net outcome from non-native insects is strongly negative. This net outcome arises because most invasive non-native insects are not directly consumed or used in any way by humans, and their overall benefits to society remain limited[24].

There are two main phenomena leading to an increased frequency of introductions and potentially expanding distributions of the costliest insect invaders: international trade[25] and global warming[9]. Invasions and subsequent expansions are exacerbated by rising human populations, movement, migration, wealth and international trade[25], despite more national and

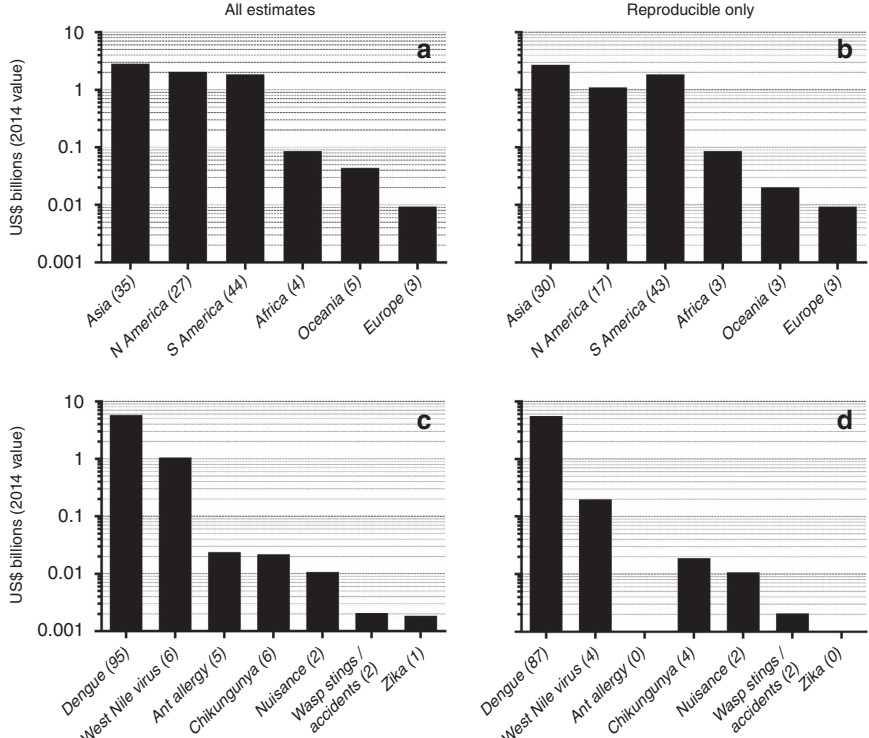

**Figure 3 | Human health costs associated with invasive insects.** Direct human health costs are categorized by major region (**a,b**) and disease (**c,d**). The first column includes all estimates regardless of reproducibility (**a,c**), whereas the second only includes costs for which estimates can be verified ('reproducible'; **b,d**). All costs expressed as annual 2014 US dollars. Bracketed numbers in the *x* axis labels indicate the number of estimates per category.

international policies targeting invasive species[26]. Climate change projections to 2050 also predict a net average increase of 18% in the area of occurrence of current arthropod invaders[27].

Given that available economic estimates are sporadic, spatially incomplete (especially outside Europe and North America), of variable reproducibility and are likely to increase as the planet warms and international trade expands, we conclude that the costs of invasive insects to human society are underestimated and will escalate with time. The available data describe only the costliest insects of mainly industrial and/or biosecurity concern, and non-market costs are rarely estimated (but see Supplementary Note 1), even though they can at times exceed market costs (for example, for forest pests[28]). In contrast, summaries of direct costs at the scale of the broader economy might not always adequately capture the true net costs of invasive insects because some investments can potentially lead to savings arising from mitigation (for example, costs of purchasing pesticides resulting in reduced damage from targeted pests). It is therefore difficult to estimate total costs from different values of direct and indirect categories of invasive insects impacts; therefore, we recommend that cost summaries always be reported by type and target (for example, Supplementary Figs 2 and 3).

Effective, early response and vigilant biosecurity are often cheaper (by up to 10 times for mosquito-borne disease[29]) than waiting to pay for accrued damages[4,9], although this might not always be the case when prevention investment occurs long before any impacts are experienced[30]. In the rare cases where those responsible for novel invasions are identified, 'polluter pays' legislation has been proposed[31]. However, most costs appear to be borne ultimately by individuals via out-of-pocket expenses[32], higher consumer prices and taxes to fund management[31], thus reinforcing the poverty-illness nexus[33]. In addition to improving guidelines for estimating the full costs of invasive insects, vigilant

planning, public-awareness campaigns and community participation could potentially relieve society of billions of dollars of annual expense, and reduce the contribution of invasive insects to human suffering.

## Methods

**Literature review.** We began our review of the literature on the economic impacts of invasive insects using the ISI Web of Science database with a specific search string to identify relevant papers (see below). We then used the Web of Science's 'refine' function to restrict the studies identified to the relevant fields, yielding 488 sources from 1911 to January 2014. We analysed each source to reject irrelevant papers and retained those containing economic estimates. We completed our database with 267 relevant studies up to December 2015 (including grey literature) opportunistically gathered. In total, we screened 737 sources, 470 of which were relevant to the economic impacts of invasive insects and from which 158 yielded useable economic estimates (Supplementary Data 1 and 2). When economic values were cited from studies not already included in the database, we searched and gathered papers, reports or chapters providing the initial estimates. For each value, we extracted the estimation methodology, and spatial and temporal coverage (full databases available in Supplementary Data 1 and 2). Owing to the diversity of the methods reviewed, we classified the reproducibility of economic values as 'reproducible' or 'irreproducible' based on qualitative criteria because of the diversity of methods reviewed (see 'Determining cost estimate reproducibility' below). We attributed 'reproducible' to values with demonstrated calculation methodologies, including uncertainties, and with available original references. 'Irreproducible' values were those without calculation methodologies, uncertainty estimates or unavailable original references (see 'Determining cost estimate reproducibility' below). We expressed all costs in 2014 US$[34]. We averaged multiple values (for example, to provide an annual average over a specified period) or uncertainty ranges before conversion to 2014 US$. In many cases we deemed some of the multiple estimates for the same invasive insect species/disease and region as redundant (that is, generally older, obsolete, incomplete or irreproducible estimates). If monetary costs were provided as a range, we used the median value for each estimate. Detailed calculations for each estimate are available in Supplementary Data 1 and 2.

For health costs, we limited the criteria for invasive vector-borne diseases and their related vector mosquitoes following Juliano and Lounibos[35], based on several life-history traits such as desiccation-resistant eggs, development in small, human-made containers, occupying human-dominated habitats, diapause and autogeny.

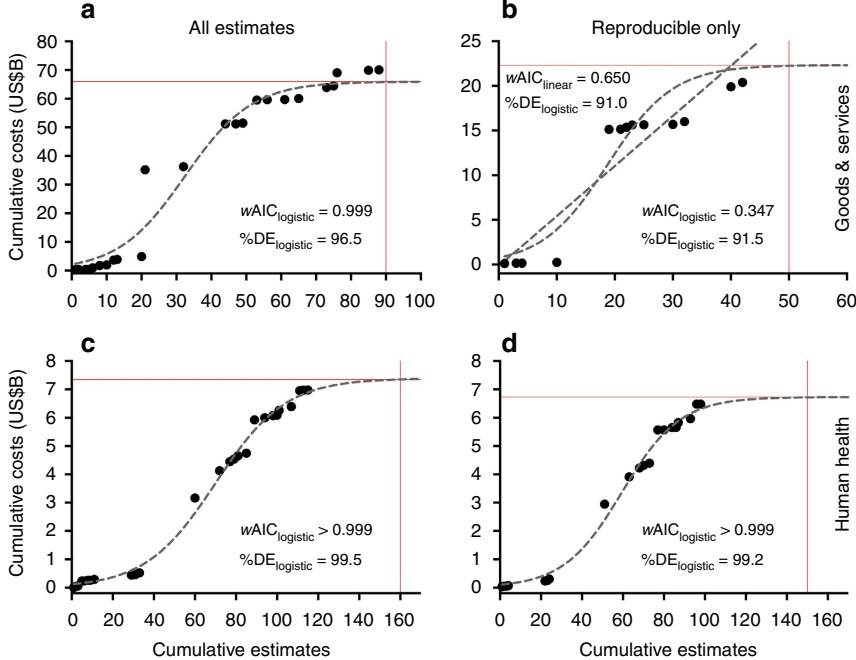

**Figure 4 | Global cumulative costs due to invasive insects.** Costs are expressed relative to the number of estimates for goods and services (**a**,**b**) and human health-related (**c**,**d**) costs, and for all estimates (**a**,**c**) and reproducible-only estimates (**b**,**d**). For a given year $t$, we summed all values (costs and number of estimates) up to $t$ (see 'Sampling bias' in the Supplementary Methods for model fitting and comparison methods). We fitted linear, exponential, logarithmic and logistic models to each curve to examine evidence for asymptotic behaviour (identified by the dominance of a logarithmic or logistic model). For all categories except reproducible-only goods and services costs (**b**), the logistic model (curvilinear grey dashed lines) had the highest Akaike's information criterion (AIC) weights ($w$AIC ≈ relative model probability) and explained >96% of the deviance in the data (%DE ≈ coefficient of determination). For reproducible-only goods and services costs (**b**), the linear model (straight grey dashed line) had the highest $w$AIC, indicating that the logistic asymptote was likely an underestimate. For each fit, we also show the approximate asymptotic cost and the associated number of cumulative estimates required to achieve the asymptote (red lines). See also Supplementary Figs 7 and 8 for accumulation curves expressed by region. All costs expressed as 2014 US dollars.

We added chikungunya and zika and excluded historical epidemics of yellow fever, dengue and malaria in South America because no estimates of these exist. Most of the economic estimates of invasive mosquito-borne diseases that we obtained concerned dengue, while only a few concerned West Nile, chikungunya and zika viruses (Fig. 3c,d), and we therefore considered the costs of these diseases to be under-represented. For this reason, we could not evaluate the many costs of epidemics (zika, chikungunya, yellow fever and dengue). Nor did we include estimates of the contribution of each disease to disability-adjusted life years (Fig. 1) because these rarely include associated financial components. To estimate annual health costs based on the outbreaks of particular diseases covering multiple years, we calculated national outbreak frequencies (annual probabilities) of disease epidemics arising from invasive insects (Supplementary Data 3).

**Search criteria for constructing the costs databases.** We searched on Web of Science in February 2014 and extracted records from 1911 to January 2014. Our search string was composed of three elements: 'invasive' AND 'insects' and 'economic impacts'. For each element we used a range of synonyms widely found in the literature. For example, for 'invasive' we used invasi*, invader, alien, exotic, non-native, introduced, naturaliz*. For 'insects', we also specified the names of a range of taxa that we identified a priori as having potentially important economic impacts. In addition, the search string included exclusion terms to reject irrelevant studies, for example, those related to medicine. We completed the search for citations in Google Scholar and internal government reports.

Full search string: TS = (invasi* OR invader OR alien OR exotic OR non-native OR introduced OR naturaliz*) AND TS = (insect* OR hymenoptera OR ant OR coleoptera OR mosquito* OR lepidoptera OR diptera OR hemiptera OR Anoplophora chinesis OR Anoplophora glabripennis OR Dendroctonus ponderosae OR Diabrotica virgifera OR Harmonia axyridis OR Leptinotarsa decemlineata OR Trogoderma granarium OR Aedes aegypti OR Aedes albopictus OR Anopheles gambiae OR Ceratitis capitata OR Culex pipiens OR Culex quinquefasciatus OR Liriomyza huidobrensis OR Aphis gossypii OR Bemisia tabaci OR Linepithema humile OR Solenopsis invicta OR Vespa velutina OR Wasmania auropunctata OR Cameraria ohridella OR Helicoverpa armigera OR Lymantria dispar OR Plutella xylostella OR Spodoptera littoralis OR Frankliniella occidentalis OR Coptotermes formosanus) AND TS = (economi* OR monetary OR dollar*)

NOT TS = (cancer* OR cardio* OR surg* OR carcin* OR engineer* OR operation OR medic* OR rotation OR ovar* OR polynom* OR purif* OR respirat* OR invasive technique).

**Removing potential double counts.** We made every effort to eliminate redundant amounts from the monetary values we used to estimate cost sums. First, we removed values that were obvious re-estimates of older values (with the more recent estimates tending to be more reproducible than older ones; for example, Supplementary Data 1, column E). We further separated costs into 'extrapolation' versus 'actual estimate' categories (columns G and H in Supplementary Data 1, respectively). Further removing those estimates already deemed irreproducible (column F), column I indicates with absolute certainty which estimates should be retained to avoid any potential case of double counting (that is, species with reproducible estimates that do not include both extrapolated and actual estimates).

The sum of estimates in column I ($22,629,029,314) versus our sum of the total costs (US$25,166,603,981) reported in the main text is only 10.1%, which suggest that even in the unlikely case of double counting, the bias is minimal, and well within the margin of error expected for a sum of median cost rates across the globe. It is essential to note that even if a species includes both extrapolations and actual values, it does not necessarily equate to double counting because often the different estimates apply to different regions of the insect's distribution or different economic components of their costs. However, this does not exclude the possibility of double counting within the irreproducible category, simply because we cannot verify how the estimates were derived to check for instances of potential double counting.

**Validity of annual cost rate metric.** It is possible that the impact rate of any invasive species will vary over time, with rates being initially low following original establishment, and then increasing as the species expands its range and possibly declining as hosts are eliminated or humans adapt to the invasion. Consequently, a simple sum of rates from many species that invaded at different points in time might not provide a practical measure of standardized costs. However, ascertaining the year of invasion of all species we examined was impossible or suspect, given a lack of monitoring data for many species. To examine the potential problem indirectly, we

plotted the cost rates versus the applicable year (median or publishing year for most goods and services estimates; initial year of reporting interval for human health estimates) for the goods and services and human health estimates separately. The subsequent bivariate plots (Supplementary Fig. 1) do not reveal any relationship with time. We therefore consider the use of cost rates as an appropriate metric for standardizing costs across species, regions and time intervals.

**Determining cost estimate reproducibility.** We determined the reliability of the cost estimates given in each study by identifying the source of all the figures used to extrapolate regional costs. When monetary values were based on available calculation methodologies, traceable original references and clearly identified uncertainties, we deemed the resulting final costs to be 'reproducible'. This reproducibility is not an assessment of quality or realism of the estimation; rather, it is a qualitative assessment of whether the initial values, assumptions and methodology applied to obtain the monetary value can be fully understood (and ideally repeated). Conversely, we defined as 'irreproducible' any monetary values that could not be fully traced, clearly understood or justified. Thus, we deemed a monetary value to be irreproducible when it was not properly referenced, was not traceable, was derived from a potentially subjective source (for example, a personal communication or a web page with no supporting references), did not have the full details of the calculations or did not provide a clear list of the underlying assumptions. We assessed reproducibility for every monetary value we found in the literature; hence, some values might be reproducible and others irreproducible in the same study (for example, ref. 36).

We could not apply the criteria in the same way to all types of monetary values. For example, assumptions and calculations are necessary when monetary values result from extrapolations (for example, see the calculations in Table 3 of ref. 37 or the values in ref. 38), but not when they are reports of raw expenses and costs (for example, values reported in ref. 39). The attribution of reproducibility was therefore a qualitative procedure specific to each monetary value. As a consequence, we supported our choices with narrative details about each value in the database (see, for example, 'detailed notes' worksheet in Supplementary Data 1).

The attribution of reproducibility to monetary estimates was clear in most cases. For example, values provided in refs 28,37,38 were explained clearly with respect to details, methodologies, assumptions and limits; therefore, we classified them as 'reproducible'. Conversely, values for *Ae. albopictus* were classified as irreproducible in ref. 6 because they were associated to a reference on *Anoplophora glabripennis*. Likewise, some values in ref. 7 were either non-sourced or were associated with personal communications, and were thus deemed irreproducible. However, in some cases, the attribution was less certain. For example, in several cases we were not able to obtain the sources of the estimates, especially for non-English sources; therefore, we conservatively decided to attribute irreproducibility to these (for example, the various values in ref. 8), although we acknowledge that they might in fact be reproducible. In the case of raw reports of expenses and costs, we generally classified values provided by official institutions as reproducible (for example, those in ref. 4), and from uncertain sources such as personal communications with no more details than the name (for example, those in ref. 8) or from conferences (for example, those in ref. 9) as irreproducible.

**Data availability.** The authors declare that all data supporting the findings of this study are available within the article and its Supplementary Information files.

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

## Acknowledgements

We thank B. Geslin and F. Chiron for discussion and direction on building the database. This work is supported in part by grants from BNP-Paribas and ANR InvaCost grants, and the Australian Research Council (FT110100306).

## Author contributions

C.J.A.B. and F.C. conceived and designed the study. B.L., M.B.-M., C.A. and A.F. compiled the database. C.J.A.B. and C.B. did the analyses. C.J.A.B. and F.C. wrote the manuscript with input from all contributing authors.

## Additional information

**Competing financial interests:** The authors declare no competing financial interests.

