## [Peer Review File · Nature Communications]

Reviewers' comments:

Reviewer #1 (Remarks to the Author):

The main objective of this paper appears to be to estimate the total economic impacts of insect invasions worldwide. This is a noble effort given that there are clearly a lot of insect invasions and many of these have vast economic impacts. The authors do a good job of pointing out that estimating these impacts is difficult and that the most widely cited previous estimates by Pimentel are deeply flawed. The authors go on to compile from the literature, estimates of impacts of several different insect species, including impacts on goods and services, control costs and impacts on human health.

The authors do not stop with estimation of total impacts. They go on to identify primary drivers of insect invasions. They also attempt to quantify the impacts of climatic warming on insect invasions and identify global hotspots of invasions. These additional efforts are probably the weakest components of the paper (see below) and detract from the central theme of economic impacts.

These issues are discussed in more detail below.

The authors' effort to estimate total economic impacts of insect invasions is noble but falls short of being either comprehensive or sophisticated. They do a nice job of "bashing" the methods used in the widely cited Pimentel papers in section 1 of the supplemental information. However, previous authors have already pointed out the shortcomings of the Pimentel estimates so the novelty of the contribution here is minimal. Furthermore, most of the examples they provide are not related to insect invasions.

The authors compile and differentiate estimates of impacts from reliable and non-reliable sources. It is valuable to differentiate these, though it seems questionable why there is any point in including the non-reliable estimates at all. The authors make an effort to differentiate economic impacts on market values, non-market values and control costs. For the purpose of summarization, they aggregate these impacts by adding them up. However, this practice may be problematic as there may be instances of "double counting" among different cost categories. This issue has been pointed out by Holmes et al, (2009) but is not discussed here.

The authors quite rightly note that they obtained cost estimates for only a tiny fraction of non-native insect species. However, they make no point to account for these "unsampled" species in deriving an estimate of total impacts. This is an important problem since the authors have data for probably less than one percent of all non-native insect species so adding up only the sampled species is clearly a vast underestimate of total impacts and makes the aggregate estimate of questionable value. Admittedly, extrapolating estimates across all species is difficult since there is likely tremendous variation in the impacts of species, with most species having zero impact. However, at least one model has been developed for extrapolating from a limited number of cost estimates across a wider pool of insect invasions. This is the model by Aukema et al (2011) which the authors cited with

regard to individual pest impact estimates but they did not address their aggregate estimates (across ~400 established non-native forest insect species established in the US). While applying the Aukema et al. model to all insect species might be a little tricky, it probably would be possible and may be much more valuable than the vast underestimate provided by simply adding up only the known examples as done here.

Another aspect of the cost estimation that is weak here is the handling of time. While the authors did report impacts here as rates (i.e. dollars / year), there are key aspects of time that need to be addressed. Specifically, the impact rate of any invasive species will likely vary considerably over time, with rates being initially low following original establishment, then increasing as the species expands its range and possibly declining as hosts are eliminated or humans adapt to the invasion. Consequently, a simple sum of rates from many species that invaded at different points in time is a slightly meaningless number. Dealing with this issue is also not a simple problem but the authors do not even mention it. Even worse, they make a rather naïve statement (182-184) that investment in prevention is always less than ultimate impacts - but this may not be the case if there is a long time period between when prevention may be conducted and when the impacts are ultimately experienced. See Olson and Roy (Olson, L. J., & Roy, S. (2002). The economics of controlling a stochastic biological invasion. *American journal of agricultural economics*, 84(5), 1311-1316.) for an explanation of how discount rates play a key role in determining whether prevention is ultimately cost effective.

Toward the end of the paper, the authors include an analysis of the drivers of insect invasions (Supplementary information, section 7). They do this by statistically relating the distribution of 5 invasive species to a host of environmental variables. First, this exercise strikes me as badly "off-topic" since it has little connection with the principal objective of the paper, namely estimation of economic impacts. Secondly, It is not clear how the analysis differs from the paper in press by Bellard et al. (51). Thirdly, the analysis is based upon the invaded range of only five species. Given that distribution data are available for a much larger number of species, it seems inexcusable to perform such an important analysis with only $n=5$. Finally, the analysis confounds at least two distinct invasion processes, which likely have very different drivers. That is the factors driving where an invasion in a continent initially occurs is likely very different from the drivers explaining where these species will ultimately spread. To make matters worse, the five species chosen by the authors include species ranging from different stages of invasion, such as *Solenopsis invicta* which has invaded and spread over a vast area and *Anoplophora glabripennis*, which currently has a very limited invaded range due to the recency of its initial invasion.

Another analysis that is inexplicably included in the manuscript is "Section 8, "projecting global invasive insect hostpots". Again, inclusion of this analysis seems like a diversion from the main point of the paper, namely economic impacts. The analysis is accomplished by using species distribution models (SDM) to predict the ultimate invaded range of 14 insect species. Once again, $n=14$ seems shockingly low given that there are many thousand species of established non-native insects in the world. The authors use their SDMs to compare total numbers of species with and without predictions of climate change and conclude from this analysis that climate change is one of the main drivers of insect

invasions. For one thing, the practice of comparing potential ranges with and without climate change is not new as there are many previous papers that have done this. But more serious is the conclusion made by the authors that climate change is one of the main drivers of biological invasions. I fail to follow their logic here. Yes, climate change can alter species distributions but the ultimate cause of the insect invasion problem is clearly globalization, more specifically trade and travel. While there may occasionally be value in examining how climate change affects insect invasions, it is clearly not the primary driver of the problem and presenting it here as such detracts from the need for policies that address their accidental movement with trade and travel.

Reviewer #2 (Remarks to the Author):

It is novel to have global estimates of the monetary costs of invasive insects. This research raise awareness of their impact both as the damage they inflict and as the cost of their management. I agree that possibly these are underestimates because the costs of many invasive species have not been quantified yet. There are missing impact type categories. For instance, introduced bees and bumblebees are infected by parasites that have jeopardized their role as crop pollinators. Other missing cost include impacts on livestock.

Other comments

Wisely the authors did not include malaria costs because the vector is not alien. However, they included cost of dengue and other human diseases. Do these values refer to costs in introduced areas, only?

Page 6, line 157-159. It is mentioned that international trade and warming are increasing the introduction of insects. Causes of other global drivers should also be included such as human movements and infrastructure development.

Page 6, l 159-160: "Introduction and subsequent invasions increase..."

Figure 1 can be simplified and should be more precise on the categories that have been included in the analysis. For instance, I doubt that monetary costs in reduction of seed dispersal or ecosystem resilience have been quantified. Moreover, terminology is redundant because they integrate several conceptual frameworks. For example, Goods and services should be included as Ecosystem services. The classification of Ecosystem Services is not standard. See as guidance Vilà et al. 2010. *Frontiers in Ecology and the Environment* 8: 135-144

Figure 2 and 3. There is no need to include "2014 value" in y-axis.

Figure 4. It is not clear how values were aggregated.

Reviewer #3 (Remarks to the Author):

There has been a lot of work go into this manuscript and it has produced an impressive survey of the literature on the global financial costs of non-native insects. Such a study does not exist elsewhere, despite the fact that this is clearly an important topic to understand in

order to protect both ecosystems and economies.

I don't doubt the general conclusion of the paper - that the costs of non-native insects are "massive yet grossly underestimated" - but I do have some concerns with the methods. My main concern is that any paper reporting any type of financial data has been included in a single pool of data to calculate the overall global annual costs. This is problematic because the data used come from a very wide range of studies that have used a very wide range of methods. *Adelges tsugae* (row 3 of source data file), for example, has been given a value of \$100million/year based on a 'willingness to pay' survey. *Agrilus plannipenis* (row 7) has a value of \$1.2462billion/yr based on an estimation of the cost of dealing with this species as it spreads across North America. *Anophlopera chinensis* (row 11) has a value of \$899k/year based on actual eradication costs. The first example is what citizens claim they would be willing to pay to prevent an invasion, the second is an estimate of what management might cost as an invader spreads, and the third is actual costs of an eradication program. These are three very different ways of addressing the financial costs of non-native species, and the values produced are not directly comparable. They should not all be thrown into a single calculation of total value. The different types of values are acknowledged by the authors because they are entered into different columns (columns E through G of source data), but these categories are not meaningfully incorporated into the actual manuscript.

The Supplementary material includes a "critique of the reliability of previous cost estimates". Four such estimates are critiqued, all of which were lead authored by David Pimentel. While I don't disagree with the critiques, it is problematic to focus on just one person's work. First, it gives the impression that all other work has been of a much higher standard (it hasn't). Second, it gives the impression of a more personal type of attack. The four publications critiqued here are based on a similar set of methods, so criticizing one of Pimentel's works is sufficient to get across the general methodological points. By going into details on four of his publications, and neglecting others, it begins to look personal.

I don't understand Figure 1. It has a lot of detail for which I could not find explanations. For example, what does the "Ease of Measurement and Direct Utility" axis refer to? What are "Costs in Common"?

As I read the manuscript I found myself being distracted by the number of analyses that are referred to briefly, but without enough detail for me to understand what was done or how I should interpret the results. I know that this is a 'short format' journal, but even considering that I think there are just too many analyses presented here. For example, in line 171-174 the idea of 'hotspots' of invasive insects is introduced, and some regions identified as such are identified. It is not clear whether these are hotspots of non-native species richness (i.e., more non-native species than other regions) or hotspots of costs (i.e., greater costs incurred). The last sentence lists some additional predicted hotspots, but the details given make it impossible to know what this means or how it was calculated.

Reviewer 1

The authors' effort to estimate total economic impacts of insect invasions is noble but falls short of being either comprehensive or sophisticated. They do a nice job of "bashing" the methods used in the widely cited Pimentel papers in section 1 of the supplemental information. However, previous authors have already pointed out the shortcomings of the Pimentel estimates so the novelty of the contribution here is minimal. Furthermore, most of the examples they provide are not related to insect invasions.

RESPONSE: We have now removed the former Section 1 given its controversy. However, it is debatable to claim that the issues with Pimentel and colleagues' work has been adequately critiqued for their shortcomings, given the sporadic and largely incomplete comments that have been published in mostly obscure journals and reports. That said, the main elements of the critique are adequately covered now in new Section 1 (formerly, Section 2) that explains how we distinguished 'reproducible' (formerly, 'reliable') from 'irreproducible' (formerly, 'unreliable') studies.

The authors compile and differentiate estimates of impacts from reliable and non-reliable sources. It is valuable to differentiate these, though it seems questionable why there is any point in including the non-reliable estimates at all.

RESPONSE: This is a good question that we debated intently. We eventually elected to keep the 'unreliable' estimates because it is possibly disingenuous and overly restrictive to exclude them entirely when they present some weaknesses. The question made us realise that we probably did not use the proper terminology between these two categories, and have now elected to change the terminology to 'reproducible' and 'irreproducible', because this nomenclature best describes what we did in our analysis. In other words, we could easily track how each study produced their estimates in the 'reproducible' category, but we could not do the same for those in the 'irreproducible' category. This updated terminology also puts a more positive slant on the distinction such that we do not unnecessarily dismiss potentially valuable estimates out of hand.

The authors make an effort to differentiate economic impacts on market values, non-market values and control costs. For the purpose of summarization, they aggregate these impacts by adding them up. However, this practice may be problematic as there may be instances of "double counting" among different cost categories. This issue has been pointed out by Holmes et al, (2009) but is not discussed here.

RESPONSE: As we originally stated in the Supplementary document ("In many cases we deemed some of the multiple estimates for the same invasive insect species/disease and region as redundant ..."), we went to considerable lengths to eliminate values from the Appendix tables that were obvious re-estimates of older values (with the more recent estimates tending to be more reproducible than the older ones). Nonetheless, we returned to our Appendix 1 Goods & Services table of values and separated them into 'extrapolation' vs. 'actual estimate' categories (columns G & H in the updated Appendix 1 Excel file). Further removing those already deemed 'redundant/archaic' (column E) and irreproducible (column F), we end up with a column (column I) indicating with absolute certainty which estimates to retain to avoid any potential case of double counting (i.e., species with reproducible estimates that do not include both extrapolated AND actual estimates).

The sum of estimates in column I (\$22,629,029,314) vs. our original sum of total costs (US\$25,166,603,981) is revealing. The difference between the two is only 10.1% of the original sum, which suggest that even in the unlikely case of double counting, the bias is minimal, and well within the margin of error expected for a sum of median annual cost rates across the globe. It is essential to note that even if a species includes both extrapolations and actual values, it does not necessarily equate to double counting because often the different estimates apply to different regions of the insect's distribution or different economic components of their total costs.

However, this does not exclude the possibility of double counting within the irreproducible category, simply because we cannot verify how the estimates were derived to check for instances of double counting. If we assume the same potential (but unconfirmed) bias of 10% from the reproducible estimates, then this potential issue is not too problematic.

As for a comparison of market and non-market costs, we only identified three non-market estimates in the goods and services total (column J), none of which was deemed archaic/redundant or irreproducible. This precludes us from making any reliable conclusion regarding the importance of non-market values in our estimates.

For the human health costs, the separation of market and non-market components does not make much sense because most estimates include a typically unspecified component of direct and indirect costs. In fact, across all health-related studies we examined, none estimated only indirect costs; 78.8% (all studies) and 92.7% (reproducible only) studies included both direct and indirect costs.

However, we have now added two new summaries of the health costs according to the expenditure type (i.e., *estimates, extrapolation, estimate+extrapolation, interview, modelling, survey, survey+estimate, unknown*) and expenditure target (i.e., *control, medical care, control+medical care, desensitization therapy, unknown*). This shows clearly that most health care costs are estimates, but there is still the issue of sampling bias (i.e., most studies [74/116] are this type of expenditure — all studies included), and that most expenditure is targeted to medical care (59/116 studies — all studies included). We have added this new information to Supplementary Section 3.

Nonetheless, we have now provided some text in new Supplementary Section 2 (*General methods*) and Section 3 (*Types and expenditure targets of health care costs*), as well as within the main text, elaborating on the issue discussed above.

The authors quite rightly note that they obtained cost estimates for only a tiny fraction of non-native insect species. However, they make no point to account for these "unsampled" species in deriving an estimate of total impacts. This is an important problem since the authors have data for probably less than one percent of all non-native insect species so adding up only the sampled species is clearly a vast underestimate of total impacts and makes the aggregate estimate of questionable value. Admittedly, extrapolating estimates across all species is difficult since there is likely tremendous variation in the impacts of species, with most species having zero impact. However, at least one model has been developed for extrapolating from a limited number of cost estimates across a wider pool of insect invasions. This is the model by Aukema et al (2011) which the authors cited with regard to individual pest impact estimates but they did not address their aggregate estimates (across ~400 established non-native forest insect species established in the US. While applying the Aukema et al. model to all insect species might be a little tricky, it probably would be possible and may be much more valuable than the vast underestimate provided by simply adding up only the known examples as done here.

RESPONSE:

While the reviewer correctly re-iterates our main message – that costs are necessarily underestimates, the contention that “... probably less than one percent ...” is merely unquantified guesswork that highlights exactly what we are avoiding in this manuscript. While we might have listed a small proportion of all the invasive insects (based on studies that have been done) imposing costs to society, we are confident that we have quantified a large proportion of the costliest species (see new Fig. 4 and new Section 5 in particular – discussed more in detail below). In fact, given their economic importance, it is highly unlikely that we have overlooked very costly species.

Our confidence is further increased by the addition of a new figure (Fig. 4) and an analysis that addresses the under-sampling question in a more sophisticated and defensible way. Previously, we extrapolated the missing costs based on a log-linear fit to regional summaries of costs and the number of associated estimates. Upon reflection, we deemed this approach was perhaps not robust to extrapolate, although the demonstration that costs increase with increasing sample size is still an essential conclusion (so we have maintained the figures in Supplementary as evidence for under-sampling). In the new Fig. 4 and Section 5 of the Supplementary Information, we show the result of different hypothetical models fitted to the cumulative costs (and number of estimates) to identify possible asymptotic behaviour. While our central tenet is maintained that under-sampling produces a gross underestimate of costs (especially for goods and services; less so for human health costs), we can now say that there is a global signal that the rate of accumulation of estimated costs is slowing, indicating that society likely estimates the costliest insects first, followed by studies that quantify costs for new areas or new invasive species. However, this evidence for deceleration is driven almost entirely by asymptotic behaviour in the well-estimated

North American goods and services costs (Fig. S5.3), whereas the slowing is more uniform across regions for human health (Fig. S5.4). It is important to note that the evidence for asymptotic cumulative costs does not necessarily imply that all costs have been adequately measured; rather, it only indicates that the costliest components of most major invasive insects have been estimated first, followed by lower costs associated with new areas becoming invaded or new insect pests being introduced to novel areas. The series of plateaus we discovered in many of the better-sampled series also indicate that there are step changes in the cumulative costs, and that future plateaus beyond the upper ones identified are still likely.

Our goal with exposing the weaknesses of Pimentel and colleagues' approach (which we have now cut back extensively following the reviewers' recommendations) was to avoid this type of unquantified guesswork. At the most, we can only say that we have vastly underestimated total costs, but we have a foundation of reasonable costs from which to improve future estimates. We have also highlighted the strong sampling bias arising from the positive association between estimate costs and the number of available estimates.

We did indeed consider applying Aukema's approach, but as the reviewer correctly identifies, this would not only be "... a little tricky ...", it would be rife with unquantifiable steps, such as estimating the number of missed species, the relative costs that these might represent, and the potential for overestimating costs for the same damage (e.g., two species of forest pest that attack the same tree species). In summary, Aukema's approach for estimating all invasive-insect costs is still completely untenable at a global scale.

Another aspect of the cost estimation that is weak here is the handling of time. While the authors did report impacts here as rates (i.e. dollars / year), there are key aspects of time that need to be addressed. Specifically, the impact rate of any invasive species will likely vary considerably over time, with rates being initially low following original establishment, then increasing as the species expands its range and possibly declining as hosts are eliminated or humans adapt to the invasion. Consequently, a simple sum of rates from many species that invaded at different points in time is a slightly meaningless number. Dealing with this issue is also not a simple problem but the authors do not even mention it.

RESPONSE: We had originally foreseen this potential problem and began to acquire information on the year of invasion to test for any relationship to the cost rate we eventually reported. However, the problem was correctly identifying the invasion date because in most instances, this was unknown or merely extrapolated. However, we recognise the oversight in not mentioning the issue and have subsequently provided an analysis that suggests that there is no time dependency in the rates we calculated. Plotting the cost rate versus the applicable year (median or publishing year for most goods and services estimates; initial year of reporting interval for human health estimates) for goods and services and human health estimates (all studies and reproducible-only studies) does not reveal any obvious relationship (see graph below):

We are therefore confident that using an annual cost rate is valid and not subject to temporal bias. We have now included this new information in Supplementary Section 2.

Even worse, they make a rather naïve statement (182-184) that investment in prevention is always less than ultimate impacts - but this may not be the case if there is a long time period between when prevention may be conducted and when the impacts are ultimately experienced. See Olson and Roy (Olson, L. J., & Roy, S. (2002). The economics of controlling a stochastic biological invasion. American journal of agricultural economics, 84(5), 1311-1316.) for an explanation of how discount rates play a key role in determining whether prevention is ultimately cost effective.

RESPONSE: We have modified the sentence accordingly:

“Effective, early response and vigilant biosecurity are often cheaper (by up to ten times for mosquito-borne disease²) than waiting to pay for accrued damages^{3,4}, although this might not be the case when prevention investment occurs long before any impacts are experienced⁵”

Toward the end of the paper, the authors include an analysis of the drivers of insect invasions (Supplementary information, section 7). They do this by statistically relating the distribution of 5 invasive species to a host of environmental variables. First, this exercise strikes me as badly "off-topic" since it has little connection with the principal objective of the paper, namely estimation of economic impacts. Secondly, It is not clear how the analysis differs from the paper in press by Bellard et al. (51). Thirdly, the analysis is based upon the invaded range of only five species. Given that distribution data are available for a much larger number of species, it seems inexcusable to perform such an important analysis with only n=5.

RESPONSE: We have removed the analysis, but have kept reference to studies suggesting drivers and mean expansion of invasive arthropods.

Finally, the analysis confounds at least two distinct invasion processes, which likely have very different drivers. That is the factors driving where an invasion in a continent initially occurs is likely very different from the drivers explaining where these species will ultimately spread. To make matters worse, the five species chosen by the authors include species ranging from

*different stages of invasion, such as *Solenopsis invicta* which has invaded and spread over a vast area and *Anoplophora glaberipennis*, which currently has a very limited invaded range due to the recency of its initial invasion.*

RESPONSE: As mentioned above, we have removed this analysis from the revised version.

Another analysis that is inexplicably included in the manuscript is "Section 8, "projecting global invasive insect hotspots". Again, inclusion of this analysis seems like a diversion from the main point of the paper, namely economic impacts. The analysis is accomplished by using species distribution models (SDM) to predict the ultimate invaded range of 14 insect species. Once again, n=14 seems shockingly low given that there are many thousand species of established non-native insects in the world. The authors use their SDMs to compare total numbers of species with and without predictions of climate change and conclude from this analysis that climate change is one of the main drivers of insect invasions. For one thing, the practice of comparing potential ranges with and without climate change is not new as there are many previous papers that have done this. But more serious is the conclusion made by the authors that climate change is one of the main drivers of biological invasions. I fail to follow their logic here. Yes, climate change can alter species distributions but the ultimate cause of the insect invasion problem is clearly globalization, more specifically trade and travel. While there may occasionally be value in examining how climate change affects insect invasions, it is clearly not the primary driver of the problem and presenting it here as such detracts from the need for policies that address their accidental movement with trade and travel.

RESPONSE: It is undeniable that climate change is a component of invasive insect spread, and we never suggested that it exceeded the importance of global trade. Regardless, we have removed the corresponding analysis from the revised manuscript.

Reviewer 2

There are missing impact type categories. For instance, introduced bees and bumblebees are infected by parasites that have jeopardized their role as crop pollinators. Other missing cost include impacts on livestock.

RESPONSE: This is a confusing statement considering that undoubtedly the most important parasites of bees and bumblebees are *Varroa destructor* mites, which are arachnids and not insects (the primary criterion of our search and topic of the manuscript). Many other insect pests of bees and bumblebees are not classified as invasive — the clear exception being invasive wasps, which we discuss in detail in the article. For livestock, we have already included all major invasive insect pests for which cost estimates exist, including the horn fly (*Haematobia irritans*), the stable fly (*Stomoxys calcitrans*), and red imported fire ants (*Solenopsis invicta*).

Further, it is essential to appreciate that we have reported all papers providing actual monetary costs, and we have not attempted to extrapolate ecological or other economically unquantifiable impacts.

Wisely the authors did not include malaria costs because the vector is not alien. However, they included cost of dengue and other human diseases. Do these values refer to costs in introduced areas, only?

RESPONSE: We can confidently claim that all dengue costs are associated with invasive vectors in each country during the period of investigation provided in each study. However, it was admittedly difficult to separate the total costs in some studies reporting them as a conglomerate of > 1 vector species.

Page 6, line 157-159. It is mentioned that international trade and warming are increasing the introduction of insects. Causes of other global drivers should also be included such as human movements and infrastructure development.

RESPONSE: 'International trade' includes human movements and infrastructure by definition. Nonetheless, we have removed the additional analyses associated with these predictions as recommended by Reviewer 1.

Page 6, l 159-160: "Introduction and subsequent invasions increase..."

RESPONSE: Changed to:

"Invasions and subsequent expansions are exacerbated by rising human populations, migration, wealth and international trade, ..."

Figure 1 can be simplified and should be more precise on the categories that have been included in the analysis. For instance, I doubt that monetary costs in reduction of seed dispersal or ecosystem resilience have been quantified. Moreover, terminology is redundant because they integrate several conceptual frameworks. For example, Goods and services should be included as Ecosystem services. The classification of Ecosystem Services is not standard. See as guidance Vilà et al. 2010. Frontiers in Ecology and the Environment 8: 135-144

RESPONSE: We admit that we struggled to construct the most informative, complete, readable and relevant figure here that we anticipated would be widely cited and reproduced if we achieved those goals. We have argued about the inclusion of *ecosystem services* and *ecological 'costs'*, and in the end decided that they are still relevant regardless of not being assessable within the aims of our manuscript. We must disagree that Millennium Ecosystem Assessment and Vilà and colleagues' all-encompassing definition^{6,7} of *ecosystem services* adequately distinguishes commercial and non-commercial 'services'; in fact, the term 'ecosystem services' itself is terminologically bankrupt due to polysemies, synonymies, disagreements on monetization aspects, and the anthropocentric nature of its intent⁸⁻¹¹. We have therefore expanded the caption to include the definition of *ecosystem services* as non-commercial, but potentially economically quantifiable to differ from the clearly *goods and services* category that is more readily monetisable.

More importantly, we have completely changed the form and appeal of the figure using a new format, colour-scheme and categorisation that we hope will please the reviewers. We have also expanded the caption to explain the meaning of our categories in more detail. We argue that it is best to keep this information in the form of a figure rather than a table because it will appeal to a broader readership and is more readable than a complex table. We also stress that a more comprehensive figure such as this is more likely to be cited, reproduced and discussed than a more restrictive figure summarising the main categories we calculated explicitly.

Figure 2 and 3. There is no need to include "2014 value" in y-axis.

RESPONSE: We disagree. The values are specific to 2014.

Figure 4. It is not clear how values were aggregated.

RESPONSE: They were simply the regional sums indicated in Fig. 2a,b and Fig. 3a,b. However, we admit we did not make this as clear as we could have in the caption of Fig. 4. Regardless, we have replaced Fig. 4 now with a more sophisticated cumulative-cost analysis, and relegated old Fig. 4 to the Supplementary Information (Fig. S5.1) and have revised its caption to:

"Sums of costs per region (regional aggregations from Fig. 2a,b and 3a,b) ..."

Reviewer 3

I don't doubt the general conclusion of the paper - that the costs of non-native insects are "massive yet grossly underestimated" - but I do have some concerns with the methods. My main concern is that any paper reporting any type of financial data has been included in a single pool of data to calculate the overall global annual costs. This is problematic because the data used come from a very wide range of studies that have used a very wide range of methods. *Adelges tsugae* (row 3 of source data file), for example, has been given a value of \$100million/year based on a 'willingness to pay' survey. *Agrilus plannipenis* (row 7) has a value of \$1.2462billion/yr based on an estimation of the cost of dealing with this species as it spreads across North America. *Anophlophera chinensis* (row 11) has a value of \$899k/year based on actual eradication costs. The first example is what citizens claim they would be willing to pay to prevent an invasion, the second is an estimate of what management might cost as an invader spreads, and the third is actual costs of an eradication program. These are three very different ways of addressing the financial costs of non-native species, and the values produced are not directly comparable. They should not all be thrown into a single calculation of total value. The different types of values are acknowledged by the authors because they are entered into different columns (columns E through G of source data), but these categories are not meaningfully incorporated into the actual manuscript.

RESPONSE: In a manner similar to the categorisation of human health cost type, we have now provided a summary of cost type for goods and services. In contrast to the result that most health costs were derived from estimates (65.5% [all]; 77.3% [reproducible only]), most goods and services costs were extrapolations based on simple to complex models (see new bottom panels of new Fig. S4.1, also reproduced below). This is despite the fact that most goods and services estimates themselves were measured (top panels, Fig. S4.1).

What can we conclude from this? Two things: (i) extrapolated (modelled) costs tend to be higher than their measured counterparts, and (ii) reproducibility favours extrapolations even more. This latter conclusion is most relevant to the reviewer's concern, for if unrealistic extrapolations dominated, we would expect that removing the irreproducible estimates would result in a lower (not higher) component of the total cost. We instead found the opposite. This goes some distance to giving us confidence that the extrapolations — while dominant in the case of goods and services totals — do not represent unrealistic estimates of real costs.

As such, we disagree that the different types of costs cannot be standardised and expressed as global figures.

Furthermore, without an attempt to summarise across all studies and estimate types, we risk diluting the overall impact of the message and the relevance of our conclusions. We feel it is essential to provide global sums as we have done, demonstrating the magnitude, uncertainties and sampling bias accordingly. Indeed, this is the only reviewer who questions

the capacity for doing so, even though we have done everything possible to avoid double-counting, to standardise, and to check assumptions.

To be clear, our message is not about providing precise cost estimates; rather, it is to demonstrate that (i) these costs are large, and (ii) that they are underestimated and under-investigated.

The Supplementary material includes a "critique of the reliability of previous cost estimates". Four such estimates are critiqued, all of which were lead authored by David Pimentel. While I don't disagree with the critiques, it is problematic to focus on just one person's work. First, it gives the impression that all other work has been of a much higher standard (it hasn't). Second, it gives the impression of a more personal type of attack. The four publications critiqued here are based on a similar set of methods, so criticizing one of Pimentel's works is sufficient to get across the general methodological points. By going into details on four of his publications, and neglecting others, it begins to look personal.

RESPONSE: We have now removed this section (see response to Reviewer 1's first comment).

I don't understand Figure 1. It has a lot of detail for which I could not find explanations. For example, what does the "Ease of Measurement and Direct Utility" axis refer to? What are "Costs in Common"?

RESPONSE: We have completely redrawn and updated Figure 1 (see specific response to Reviewer 2 above) to make it easier to understand and more relevant to our manuscript's main themes.

As I read the manuscript I found myself being distracted by the number of analyses that are referred to briefly, but without enough detail for me to understand what was done or how I should interpret the results. I know that this is a 'short format' journal, but even considering that I think there are just too many analyses presented here. For example, in line 171-174 the idea of 'hotspots' of invasive insects is introduced, and some regions identified as such are identified. It is not clear whether these are hotspots of non-native species richness (i.e., more non-native species than other regions) or hotspots of costs (i.e., greater costs incurred). The last sentence lists some additional predicted hotspots, but the details given make it impossible to know what this means or how it was calculated.

RESPONSE: We have now removed the hotspots analysis from the Supplementary Information.

References

- 1 Su, N. Y. Novel technologies for subterranean termite control. *Sociobiology*. **39**, 1-7 (2002).
- 2 Vazquez-Prokopec, G. M., Chaves, L. F., Ritchie, S. A., Davis, J. & Kitron, U. Unforeseen costs of cutting mosquito surveillance budgets. *PLoS Negl. Trop. Dis.* **4**, e858, doi:10.1371/journal.pntd.0000858 (2010).
- 3 Aukema, J. E. *et al.* Historical accumulation of nonindigenous forest pests in the continental United States. *BioScience* **60**, 886-897, doi:10.1525/bio.2010.60.11.5 (2010).
- 4 Bebbler, D. P., Ramotowski, M. A. T. & Gurr, S. J. Crop pests and pathogens move polewards in a warming world. *Nat. Clim. Change*, 1-4, doi:10.1038/NCLIMATE1990 (2013).
- 5 Olson, L. J. & Roy, S. The economics of controlling a stochastic biological invasion. *Am. J. Agric. Econ.* **84**, 1311-1316 (2002).
- 6 Millennium Ecosystem Assessment. *Ecosystems and Human Well-being: Synthesis*. (Island Press, 2005).
- 7 Vilà, M. *et al.* How well do we understand the impacts of alien species on ecosystem services? A pan-European, cross-taxa assessment. *Front. Ecol. Environ.* **8**, 135-144, doi:10.1890/080083 (2010).
- 8 Boyd, J. & Banzhaf, S. What are ecosystem services? The need for standardized environmental accounting units. *Ecol. Econ.* **63**, 616-626, doi:http://dx.doi.org/10.1016/j.ecolecon.2007.01.002 (2007).
- 9 Swift, M. J., Izac, A. M. N. & van Noordwijk, M. Biodiversity and ecosystem services in agricultural landscapes—are we asking the right questions? *Agriculture, Ecosystems & Environment* **104**, 113-134, doi:http://dx.doi.org/10.1016/j.agee.2004.01.013 (2004).
- 10 Spangenberg, J. H. & Settele, J. Precisely incorrect? Monetising the value of ecosystem services. *Ecol. Complex* **7**, 327-337, doi:http://dx.doi.org/10.1016/j.ecocom.2010.04.007 (2010).

- 11 Chan, K. M. A., Satterfield, T. & Goldstein, J. Rethinking ecosystem services to better address and navigate cultural values. *Ecol. Econ.* **74**, 8-18, doi:<http://dx.doi.org/10.1016/j.ecolecon.2011.11.011> (2012).

REVIEWERS' COMMENTS:

Reviewer #2 (Remarks to the Author):

I'm happy with the responses provided except for Figure 1 which it is not eye catching, too complex and mixes different terminologies and schools of thought. Eg. "services" is repeated at different levels of circles making it confusing. I suggest to substitute it by a figure (or table) including detailed cost categories and subcategories considered in this study.

Reviewer #3 (Remarks to the Author):

Review of NCOMMS-16-03682A

I was Reviewer Three in the first round of reviews.

Overall this is an improved submission. I found it easier to follow now that there are fewer analyses. What is included is generally better explained.

I will offer comments on what I see as some of the biggest changes, and on the responses to my first review.

Figure 1: It is now possible to follow this figure, but it is complex and it took me several minutes to wrap my head around the various axes of information.

Figure 2: Good.

Figure 3: Good.

Figure 4: I have read and re-read the text in the manuscript describing this analysis, the text in the supplementary material, and the text in figure description itself. I don't understand how this analysis was conducted or how the outcomes should be interpreted.

Responses to my Review:

Comment 1: The response to my comment misses the point that I was trying to make (my apologies if it was not sufficiently clear). My point is that many of the costs included actually measure very different things. The 'willingness to pay' measure, for example, is not a measure of cost, but a measure of how much people fear an invasion that may happen in the future. It bears little relation to the actual cost of addressing the invasion should it occur. To give an example, my willingness to pay to avoid breaking my arm bears no relation to the financial cost I would incur should I actually break my arm. In fact, I have no idea how much it would cost. Thus, my expression of willingness to pay is not an estimate of the financial cost that would occur, but is a measure of how much I fear that type of injury.

I know of three main criticisms of Pimentel's work, and of other work like his. I think that this paper addresses one of them.

The first criticism is that Pimentel takes limited data (e.g., an estimate of cost of an invaders per unit area in one state) and extrapolates them widely (e.g., assumes those costs will be the same per unit area across many states). The present manuscript addresses this criticism.

The second criticism is that Pimentel lumps together any available cost estimates (essentially anything with a dollar sign) and does not adequately deal with the fact that they are measuring different things. See my comments above for why I think the present paper does not address this criticism.

The third criticism is that Pimentel considers that any costs occur in a vacuum and not within a wider economy. As an example, some of the costs of invasive insects come from the need to purchase pesticides. While those pesticides may be expensive, purchasing them produces benefits for the wider economy by supporting other industries. Thus, expenditures on control/mitigation are not a complete loss to the economy. I know of several undergraduates who have worked as interns addressing invasive species. The present study (and Pimentel) would count their salaries as a cost, but doesn't consider the benefits that those students receive. I am not arguing that this final criticism can be fully addressed in this study (or in any study like it), but it should be acknowledged that, at the scale of the broader economy, the net costs of invasive insects may in fact be lower than is estimated when only the direct costs are considered.

Reviewer 2

I'm happy with the responses provided except for Figure 1 which it is not eye catching, too complex and mixes different terminologies and schools of thought. Eg. "services" is repeated at different levels of circles making it confusing. I suggest to substitute it by a figure (or table) including detailed cost categories and subcategories considered in this study..

RESPONSE: We now refer only once to 'services' in the main category of 'goods and services'; we have rephrased the other use of 'regulating services' to 'ecosystem processes'. See the more elaborate response to the Editor above.

Reviewer 3

Figure 1: It is now possible to follow this figure, but it is complex and it took me several minutes to wrap my head around the various axes of information.

RESPONSE: See responses above.

Figure 4: I have read and re-read the text in the manuscript describing this analysis, the text in the supplementary material, and the text in figure description itself. I don't understand how this analysis was conducted or how the outcomes should be interpreted.

RESPONSE: We are not certain we could be much clearer here, and perhaps the confusion stems from the reviewer's possible unfamiliarity with parsimony scores of model performance. In essence, we fit several models each linked to a specific hypotheses (e.g., a linear or exponential rise in cumulative costs not indicative of asymptotic behaviour, whereas a logistic model indicates asymptotic behaviour). We then calculated their relative model probabilities using Akaike's information criterion (AIC) weights, now a standard practice in ecology and evolution¹⁻⁵. The models with the lowest AIC (i.e., highest model weights) indicate the relative probability of the model fitting the data *after incorporating a penalty for the number of parameters included in each model*. Likewise, the percent deviance explained provides a measure of how much variance in the data is explained by the model (analogous to a least-squares R^2 value). We have now added a little more detail in the Supplementary Results and Methods (*Sampling bias* section) regarding these aspect, as well as some clarification in the legend of Fig. 4.

The response to my comment misses the point that I was trying to make (my apologies if it was not sufficiently clear). My point is that many of the costs included actually measure very different things. The 'willingness to pay' measure, for example, is not a measure of cost, but a measure of how much people fear an invasion that may happen in the future. It bears little relation to the actual cost of addressing the invasion should it occur. To give an example, my willingness to pay to avoid breaking my arm bears no relation to the financial cost I would incur should I actually break my arm. In fact, I have no idea how much it would cost. Thus, my expression of willingness to pay is not an estimate of the financial cost that would occur, but is a measure of how much I fear that type of injury.

RESPONSE: We agree and do not dispute these sentiments. However, after re-examining our calculations for the presence of 'willingness to pay' estimates, we identified only 3 studies of two species:

- Holmes *et al.*⁶ (*Adelges tsugae*)
- Leuschner *et al.*⁷ (*Lymantria dispar*)
- Sills⁸ (*Lymantria dispar*)

In all three cases, we had already classified these studies as redundant/archaic, so their estimates that include willingness to pay were not incorporated into the final summaries. For both species listed, we instead used primarily the values included in Aukema *et al.*⁹, who did not include non-market values.

The first criticism is that Pimentel takes limited data (e.g., an estimate of cost of an invaders per unit area in one state) and extrapolates them widely (e.g., assumes those costs will be the same per unit area across many states). The present manuscript addresses this criticism.

RESPONSE: Thank you. We have now added the following clause to the first sentence of the new summary paragraph at the end of the Discussion to reflect this idea:

“Taking all reported goods and services estimates, and avoiding the extrapolation of limited data, invasive insects cost ...”

The second criticism is that Pimentel lumps together any available cost estimates (essentially anything with a dollar sign) and does not adequately deal with the fact that they are measuring different things. See my comments above for why I think the present paper does not address this criticism.

RESPONSE: Although we have averted the potential problem of incorporating willingness-to-pay costs (see response to the comment directly above), we agree that total sums can be partially misleading if the constituents are from vastly different categories. It is for this reasons we have provided the exhaustive breakdown of the different cost types and categories in the supplementary material as requested in the previous revision. We have therefore added the following sentences to the penultimate paragraph of the Discussion:

“In contrast, summaries of direct costs at the scale of the broader economy might not always adequately capture the true net costs of invasive insects because some investments can potentially lead to savings arising from mitigation (e.g., costs of purchasing pesticides resulting in reduced damage from targeted pests). It is therefore difficult to estimate total costs from different values of direct and indirect categories of invasive insects impacts, so we recommend that cost summaries always be reported by type and target (e.g., Supplementary Figs. 2 and 3).”

The third criticism is that Pimentel considers that any costs occur in a vacuum and not within a wider economy. As an example, some of the costs of invasive insects come from the need to purchase pesticides. While those pesticides may be expensive, purchasing them produces benefits for the wider economy by supporting other industries. Thus, expenditures on control/mitigation are not a complete loss to the economy. I know of several undergraduates who have worked as interns addressing invasive species. The present study (and Pimentel) would count their salaries as a cost, but doesn't consider the benefits that those students receive. I am not arguing that this final criticism can be fully addressed in this study (or in any study like it), but it should be acknowledged that, at the scale of the broader economy, the net costs of invasive insects may in fact be lower than is estimated when only the direct costs are considered.

RESPONSE: While we are of the opinion that the underestimates of total costs vastly outweigh this possibility, we agree that it is worth mentioning. We have now added the following to the penultimate paragraph of the Discussion:

“In contrast, summaries of direct costs at the scale of the broader economy might not always adequately capture the true net costs of invasive insects because some investments can potentially lead to savings arising from mitigation (e.g., costs of purchasing pesticides resulting in reduced damage from targeted pests).”

References

- 1 Burnham, K. P. & Anderson, D. R. Kullback-Leibler information as a basis for strong inference in ecological studies *Wildl. Res.* **28**, 111-119 (2001).
- 2 Burnham, K. P. & Anderson, D. R. *Model Selection and Multimodel Inference: A Practical Information-Theoretic Approach*. 2nd edn, (Springer-Verlag, 2002).
- 3 Burnham, K. P. & Anderson, D. R. Understanding AIC and BIC in model selection *Sociological Methods and Research* **33**, 261-304 (2004).
- 4 Link, W. A. & Barker, R. J. Model weights and the foundations of multimodel inference *Ecology* **87**, 2626-2635, doi:10.1890/0012-9658(2006)87[2626:MWATFO]2.0.CO;2 (2006).
- 5 Elliott, L. P. & Brook, B. W. Revisiting Chamberlain: multiple working hypotheses for the 21st Century *BioScience* **57**, 608-614 (2007).

- 6 Holmes, T. P., Aukema, J. E., Von Holle, B., Liebhold, A. & Sills, E. Economic impacts of invasive species in forests *Ann. N. Y. Acad. Sci.* **1162**, 18-38, doi:10.1111/j.1749-6632.2009.04446.x (2009).
- 7 Leuschner, W. A., Young, J. A., Waldon, S. A. & Ravlin, F. W. Potential benefits of slowing the gypsy moth's spread *South. J. Appl. For.* **20**, 65-73 (1996).
- 8 Sills, E. Assessment of the Economic Feasibility of the Gypsy Moth Slow the Spread Project. 72 (Final Report to USDA Forest Service State & Private Forestry Grant #NC-06-DG-11244225-337, Raleigh, North Carolina, 2008).
- 9 Aukema, J. E. *et al.* Economic impacts of non-native forest insects in the continental United States *PLoS One* **6**, e24587-e24587, doi:10.1371/journal.pone.0024587 (2011).